# Isotherm and Kinetic Study of Metal Ions Sorption on Mustard Waste Biomass Functionalized with Polymeric Thiocarbamate

**DOI:** 10.3390/polym15102301

**Published:** 2023-05-13

**Authors:** Dumitru Bulgariu, Lăcrămioara (Negrilă) Nemeş, Iftikhar Ahmad, Laura Bulgariu

**Affiliations:** 1Department of Geology, Faculty of Geography and Geology, “Alexandru Ioan Cuza” University of Iaşi, 700050 Iaşi, Romania; dbulgariu@yahoo.com; 2Romanian Academy, Filial of Iaşi, Branch of Geography, 700050 Iaşi, Romania; 3Department of Environmental Engineering and Management, “Cristofor Simionescu” Faculty of Chemical Engineering and Environmental Protection, Technical University Gheorghe Asachi of Iasi, 700050 Iaşi, Romania; nemeslacramioara@yahoo.com; 4Department of Environmental Sciences, COMSATS University Islamabad, Vehari 61100, Pakistan; iftikharahmad@cuivehari.edu.pk

**Keywords:** sorption, metal ions, mustard waste biomass, functionalization, polymeric thiocarbamate

## Abstract

The presence of high concentrations of metal ions in effluents resulting from industrial metal coatings is a well-known fact. Most of the time, such metal ions, once they reach the environment, significantly contribute to its degradation. Therefore, it is essential that the concentration of metal ions is reduced (as much as possible) before such effluents are discharged into the environment to minimize the negative impact on the quality of the ecosystems. Among all methods that can be used to reduce the concentration of metal ions, sorption is one of the most viable options due to its high efficiency and low cost. Moreover, due to the fact that many industrial wastes have sorbent properties, this method is in accordance with the principles of circular economy. Based on these considerations, in this study, mustard waste biomass (resulting from oil extraction) was functionalized with an industrial polymeric thiocarbamate (METALSORB) and used as a sorbent to remove Cu(II), Zn(II) and Co(II) ions from aqueous media. The best conditions for the functionalization of mustard waste biomass were found to be: mixing ratio biomass: METASORB = 1 g: 1.0 mL and a temperature of 30 °C. The experimental sorption capacities of functionalized sorbent (MET-MWB) were 0.42 mmol/g for Cu(II), 0.29 mmol/g for Zn(II) and 0.47 mmol/g for Co(II), which were obtained under the following conditions: pH of 5.0, 5.0 g sorbent/L and a temperature of 21 °C. The modeling of isotherms and kinetic curves as well as the analysis of the results obtained from desorption processes demonstrate the usefulness of this sorbent in the treatment of effluents contaminated with metal ions. In addition, tests on real wastewater samples highlight the potential of MET-MWB for large-scale applications.

## 1. Introduction

Environmental pollution resulting from the intensification of industrial activities is, by far, one of the most important problems facing many countries around the world. Thus, the inadequate disposal of production waste, industrial effluents or gaseous emissions lead to the accumulation of dangerous pollutants in the environment (water, air, soil), causing the degradation of the ecosystems. That is why it is unanimously accepted that treating industrial emissions before they are discharged into the environment is the most effective way to reduce environmental pollution [1,2,3]. This strategy has the widest applicability in the case of liquid effluents (washing water, process water, wastewater, etc.), because: (i) their transport is expensive, even over short distances; (ii) their storage is difficult and requires special precautions; and (iii) they may contain a wide variety of pollutants (organic compounds and/or metal ions) in a wide range of concentrations, which can react over time. Thus, for liquid effluents, various physical, physico-chemical and chemical methods for pollutant removal have been developed, which take into account the nature of the pollutants, their concentration and the volume of effluent to be treated [4,5,6]. In the case of industrial effluents containing metal ions, methods such as chemical precipitation, ion exchange, coagulation–flocculation, osmosis, etc. [6,7,8,9], have proven their applicability on a large scale. All of these methods have been successfully applied to reduce the content of metal ions in industrial effluents, and the main criterion for selecting a specific method is their removal efficiency [10]. This is because it is well known that most metal ions that have a high toxic potential (even at low concentrations) are non-biodegradable (so they do not break down over time), have a strong tendency to accumulate in the environment and significantly affect the quality of ecosystems and human life [11,12]. Thus, numerous disorders of the nervous system, including cardiovascular, endocrine, pulmonary, etc., have been proven to be determined by the presence of metal ions in drinking water or in plants [13,14] as a consequence of environmental pollution. Therefore, it is essential to remove metal ions from industrial effluents before they are discharged into the environment, and currently, there are strict legislative regulations in this regard [14]. 

However, in the current context of the circular economy, the removal efficiency of metal ions can no longer be considered the only criterion for the selection of the treatment method for industrial effluents. Cost, energy consumption and the amount of secondary waste are also aspects to consider when choosing a sustainable method. Unfortunately, most previously mentioned methods of treating industrial effluents containing metal ions are high-energy consumers, require high expenses and generate significant amounts of secondary waste, which in turn need to be treated [15]. Therefore, finding a metal ion removal method in which these disadvantages are minimized is a challenge for research in this field.

Compared to traditional methods, sorption is considered a much more ecological method that can be used to remove metal ions from aqueous media (including industrial effluents) [16,17,18,19], because: (i) it is low cost and simple to operate; (ii) it can be easily adapted to treat both small and large volume of effluent; (iii) the sorbent used can be easily regenerated and the retained metal ions can be recovered; and (iv) it allows the valorization of a wide range of agricultural and industrial waste, which can be used as sorbents. 

However, the applicability of large-scale sorption processes is quite limited. This is because the percentage of metal ion removal hardly reaches 95–97%, which is much lower compared to chemical precipitation, where the removal percentage often exceeds 99% [16]. This difference is mainly due to the limited number of active centers on the surface of most sorbents, which can interact with metal ions in aqueous media. To solve this problem, many studies in the literature propose the functionalization of cheap and widely available materials with organic reagents [20,21,22,23] in order to obtain sorbents with high efficiency for the treatment of industrial effluents. 

Two aspects must be taken into account in the design of such a sorbent. The first is related to the choice of solid material to be functionalized. Such material must be available in large quantities, simple to prepare and stable over time. Biomass waste from the bio-fuels industry meets these requirements and also have no other uses compared to natural materials or other agricultural waste. In addition, since such biomass wastes result from a the solvent extraction step, they are already ground and sieved; therefore, the preparation cost is significantly reduced. Moreover, due to the breaking of the cell walls during the extraction step [24], their specific surface area is high (compared to the initial biomass), which is a major advantage in the subsequent functionalization step. Based on these considerations, mustard waste biomass (MWB) has been selected for experimental studies, because: (i) mustard biomass is frequently used on the industrial scale for oil extraction, due to its high content (over 39%) [25]; (ii) the breaking of the cell walls (due to the extraction of oil) increased the specific surface area and the number of superficial functional groups [26], which makes this biomass waste suitable for functionalization with organic reagents; and (iii) this waste has no practical uses other than incineration (due to the traces of organic solvents).

The second aspect is related to the type of organic reagent used for functionalization. Such organic reagents must be easy to prepare, without requiring laborious synthesis procedures (which would increase the cost of functionalization), be non-toxic and have adequate chemical stability. In addition, their functional groups must be spatially available and have affinity for metal ions in aqueous solution. Many organic reagents fulfil these requirements and have been used in the literature for functionalization [21,27]. However, our attention turned to METALSORB. METALSORB is a polymeric thiocarbamate that is used on an industrial scale as a precipitating agent for the removal of metal ions from aqueous media. Due to its high solubility in water and the large number of functional groups with N and S donor atoms, this organic reagent allows the quantitative removal (>99.5%) of a large number of metal ions (Cu(II), Zn(II), Ni(II), Pb(II), Hg(II), etc.) [28]. However, the use of METASORB in the treatment of industrial effluents by precipitation has two major disadvantages, namely (i) the excess of the organic reagent added for precipitation (which does not react with metal ions) leads to the contamination of the effluents with organic compounds, and therefore, to an increase in the value of chemical oxygen demand (COD, mg O_2_/L); and (ii) the recovery of metal ions from the precipitate is difficult (due to the high stability of the chelated complexes), which is why the sludge obtained after the removal of metal ions is stored for destruction.

In order to minimize these disadvantages, in this study, METALSORB was used for the functionalization of mustard waste biomass. The obtained material (MET-MWB) was then tested as a sorbent for the removal of Cu(II), Zn(II) and Co(II) ions from aqueous media. The experiments were performed in batch systems, as a function of the initial metal ion concentration and contact time, both in the laboratory solutions and in real wastewater samples. Special attention was paid to the optimization of the functionalization conditions and the characterization of the obtained sorbent. The structural characteristics of MET-MWB and the detailed analysis of its sorptive performance highlight the applicative potential of this environmentally friendly sorbent in metal ion removal processes. 

## 2. Materials and Methods

### 2.1. Chemical Reagents

All chemical reagents used in this study were of analytical grade and were purchased from the Chemical Company (Iaşi, Romania). The stock solutions of metal ions (10^−2^ mol/L) were prepared by dissolving appropriate an amount of metal sulphate salts (CuSO_4_, ZnSO_4_ and CoSO_4_) in distilled water. Volumes from the stock solutions were diluted with distilled water in order to prepare the working solutions at different concentrations. In each case, the pH of the working solutions was adjusted by using HNO_3_ or NaOH solutions (10^−2^ mol/L) and measured by a pH/ion-meter (MM743 type, equipped with a combined glass electrode). 

METALSORB was purchased from FLochem Industries, and was used as received. This polymeric thiocarbamate (Figure 1) is an orange liquid, with a density of 1.1613 g/cm^3^ and an average molecular weight of 5000 g/mol. Due to these characteristics, the required volumes of METALSORB were measured with a pipette. 

### 2.2. Preparation and Characterization of Sorbent

Mustard seeds purchased from a local farm (Iaşi, Romania) were washed several times with distilled water (to remove solid impurities and dust), air-dried (at room temperature), ground and sieved. The mustard biomass was subjected to an oil extraction stage (n-hexane, 36 h, Soxhlet extractor) to obtain mustard waste biomass (MWB). After air-drying (at ambient temperature), MWB was mortared (for homogenization) and mixed with METALSORB in different proportions (1 g: 0.5 mL; 1 g: 1 mL; and 1 g: 2 mL) and at different temperatures (20–40 °C). The choice of this temperature range was made taking into account the thermal stability of METASORB specified in the technical data sheet. Each sample was stirred (at 150 rpm) in a thermostatic water bath for 3 h. After filtration (through quantitative filter paper), the sorbents were washed three times with distilled water, dried at 50 °C for 6 h (to remove humidity) and kept in desiccators. The selection of the most efficient sorbent for the experimental studies was performed by testing the sorption capacity of each material for Cu(II) ions, according to the methodology described in Section 2.3. 

The characterization of the sorbent (MET-MWB) used in the sorption experiments was carried out by (i) SEM microscopy (SEM Hitachi S3000N (Tokyo, Japan), 20 kV), which allows the morphological analysis of the sorbent surface; and (ii) FTIR spectrometry (Bio-Rad Spectrometer, spectral range of 400–4000 cm^−1^, resolution of 4 cm^−1^, KBr pellet technique) to identify the nature of the superficial functional groups of the sorbent. 

### 2.3. Sorption/Desorption Studies

To examine the efficiency of MET-MWB in metal ion removal processes, the sorption experiments were conducted in batch systems for each metal ion (mono-component systems) as a function of their initial concentration and contact time. Thus, 25 mL of metal ion solution (Cu(II), Zn(II) and Co(II)) with an initial solution pH of 5.0 were mixed with 0.125 g MET-MWB at room temperature (21 ± 1 °C). The values of the initial solution’s pH (5.0), sorbent dose (5 g/L) and temperature (21 °C) were established as optimal in a previous study [29]. To study the influence of the initial metal ion concentration on the sorption efficiency, experimental isotherms were recorded, where the initial concentration of metal ions ranging from 0.2 to 3.2 mmol/L. In these experiments, the solution pH (5.0), sorbent dose (5 g/L), contact time (24 h) and temperature (21 °C) were kept constant. The effect of the contact time was examined in the range of 5–180 min, for a concentration of metal ions of 0.8 mmol/L, while the other parameters were constant (as mentioned above). After phase separation (by filtration on quantitative filter paper), the metal ion concentration was determined by atomic absorption spectrometry (AAS NovAA 400 P Spectrometer, acetylene/air flame, using a prepared calibration graph), and the sorption parameters were calculated using the relations:(1)$$\mathrm{Sorption}\;\mathrm{capacity}:\;q=\frac{(c_{0}-c)\cdot V}{m}$$
(2)$$\mathrm{Removal}\;\mathrm{percent}:\;R=\frac{c_{0}-c}{c_{0}}\cdot 100$$
where *c_o_* and *c* are the initial and final concentrations of metal ions (mmol/L); *V* is the volume of the solution (L) and *m* is the mass of sorbent used in each experiment (g).

In desorption experiments, 0.25 g of MET-MWB loaded with each metal ion was mixed with 10 mL of HNO_3_ (10^−1^ mol/L) and stirred for 3 h. After filtration, the metal ion concentration was analyzed as mentioned above. The regenerated sorbent was then washed several times with distilled water (until reaching a neutral pH) and used in another sorption cycle. Three sorption/desorption cycles were performed for each metal ion and the regeneration efficiency (RE, %) of MET-MWB was calculated using the relation:(3)$$\mathrm{RE}=\frac{q_{r}}{q_{0}}\cdot 100$$
where q_0_ is the sorption capacity before regeneration (mmol/g) and q_r_ is the sorption capacity after regeneration (mmol/g).

Duplicate experiments were performed for each sorption and desorption series, and the mean value of the experimental results was used for calculations and in the graphical representations. All results were analyzed by ANOVA and *p*-values of less than 0.05 were considered significant.

### 2.4. Isotherm and Kinetic Modeling

The experimental isotherms obtained for the sorption of Cu(II), Zn(II) and Co(II) on MET-MWB were analyzed using three isotherm models, namely the Langmuir model, Freundlich model and Temkin model, whose mathematical equations [30,31], respectively, are:(4)$$\mathrm{Langmuir}\;\mathrm{model}:\;\frac{1}{q}=\frac{1}{q_{\max}\cdot K_{L}}\cdot \frac{1}{c}$$
(5)$$\mathrm{Freundlich}\;\mathrm{model}:\;\log q=\log K_{F}+\frac{1}{n}\cdot \log c$$
(6)$$\mathrm{Temkin}\;\mathrm{model}:\;q=B\ln A_{T}+B\ln c$$
where *q* is the sorption capacity, (mmol/g); *q_max_* is maximum sorption capacity, (mmol/g); *K_L_* is the Langmuir constant, (L/g); *c* is the equilibrium concentration of the metal ions, (mmol/L); *K_F_* is the Freundlich constant, (L/g); *n* is the heterogeneity factor; *A_T_* is the equilibrium binding constant, (L/g); and *B* is the constant correlated with the heat of sorption, (J/mol).

The selection of these three isotherm models was made taking into account the assumptions underlying their theoretical foundation [31], allowing one to establish the way in which the metal ions bind to the surface of the sorbent and the nature of the interaction that takes place during the sorption process.

The modeling of the experimental kinetic data was performed using pseudo-first-order, pseudo-second-order and intra-particle diffusion kinetic models. The mathematical equations of these models [32,33,34] are:(7)$$\mathrm{Pseudo}\text{-}\mathrm{first}\text{-}\mathrm{order}\;\mathrm{model}:\;\log(q_{e}-q_{t})=\log q_{e}-k_{1}\cdot t$$
(8)$$\mathrm{Pseudo}\text{-}\sec\mathrm{ond}\text{-}\mathrm{order}\;\mathrm{model}:\;\frac{t}{q_{t}}=\frac{1}{k_{2}\cdot q_{e}^{2}}+\frac{t}{q_{e}}$$
(9)$$\mathrm{Intra}\text{-}\mathrm{diffusion}\;\mathrm{particle}\;\mathrm{model}:\;q_{t}=k_{diff}\cdot t^{1/2}+c$$
where *q_e_*, *q_t_* are the sorption capacity at equilibrium and at different *t* values, (mmol/g); *k*_1_ is the rate constant of the pseudo-first-order model, (1/min); *k*_2_ is the rate constant of the pseudo-second-order model, (g/mmol min); *k_diff_* is the intra-particle diffusion rate constant, (mmol/g min^1/2^); and *c* is the concentration of metal ions, (mmol/L).

These kinetic models were chosen because they are useful to determine the rate and order of the sorption process, and thus allow the evaluation of MET-MWB performances.

The most appropriate models (isotherm and kinetic) for the description of the experimental data were evaluated based on the value of the regression coefficients (R^2^) calculated from the statistical analysis.

### 2.5. Real Industrial Effluents Tests

An industrial wastewater sample obtained from a local metal coating company was used to test the efficiency of the MET-MWB biosorbent to remove Cu(II), Zn(II) and Co(II) ions from real samples. Before use, the wastewater sample was filtered (on quantitative filter paper) to remove solid impurities and conditioned (24 h) at room temperature. One hundred milliliters of industrial wastewater was used for the sorption experiments of each metal ion using the MET-MWB sorbent. The initial concentration of each metal ion was adjusted to 0.8 mmol/L and the initial pH to 5.0. Each liquid phase was added over 0.5 g of sorbent and mixed for 3 h (at 21 °C). After filtration, the concentration of metal ions in the solution was determined as described above (AAS spectrometry, using a prepared calibration graph), while the standard methods [35] were used to determine the other quality parameters of the effluent (before and after metal ion sorption).

## 3. Results and Discussion

As mentioned previously, the main disadvantage of biomass resulting from oil extraction (such as MWB) is the traces of organic solvent that remain in the biomass composition, even after several washing steps. This significantly limits the possibilities of using MWB for known applications (such as animal feed or bedding, soil compost, etc.) [25,29], but opens new opportunities for increasing the efficiency of this material in sorption processes through functionalization. METALSORB was selected for the functionalization of MWB because (i) it is highly efficient in the removal of metal ions by precipitation, being used on an industrial scale; and (ii) the compatibility between the hydrocarbon radicals in the METALSORB molecule (Figure 1) and traces of organic solvent in MWB allow one to obtain a stable functionalized sorbent over time.

Therefore, in order to highlight the efficiency of the functionalized sorbent in the metal ion removal process, it is necessary to take into account: (i) the establishment of the optimal functionalization conditions; (ii) the structural characterization of the functionalized sorbent; and (iii) its testing in the sorption processes, both for laboratory solutions and in real samples.

### 3.1. Establishing the Optimal Functionalization Conditions

To establish the optimal functionalization conditions, two experimental parameters were considered, namely the volume of METALSORB solution used for functionalization and temperature. In each case, the efficiency of the obtained sorbent was quantitatively evaluated using its sorption capacity for Cu(II) ions (see Section 2.2), and the obtained experimental results are illustrated in Figure 2.

It can be seen that both experimental parameters influence the efficiency of the functionalization process (Figure 2). Thus, if in the case of the volume of the METALSORB solution, the increase in its value causes an increase in the sorption capacity by more than 90% compared to non-functionalized MWB (Figure 2a), in the case of temperature, the variation in the sorption capacity is much smaller in the studied range (20.76%) (Figure 2b).

The increase in the sorption capacity for Cu(II) ions when treating MWB with METASORB shows that the functionalization process was successfully carried out. Moreover, with the increase in the volume of METASORB solution used for functionalization, the number of superficial functional groups of sorbent also increases, which favors the more effective retention of Cu(II) ions from the aqueous solution. However, since the difference between the sorption capacities in the case of treating MWB with 1.0 and 2.0 mL of METALSORB is below 3%, 1.0 mL of METASORB solution was considered sufficient for the functionalization of 1 g of MWB, and this value was selected as optimal (Figure 2a). Selecting this value as optimal for MWB functionalization allows one to both obtain an efficient sorbent for the removal of metal ions and maintain a low cost of sorbent preparation low.

On the other hand, increasing the temperature has the effect of both increasing the thermal agitation and decreasing of the viscosity of the METALSORB solution. Consequently, the efficiency of the functionalization process increases, leading to an increase in the sorption capacity for Cu(II) ions (Figure 2b). However, in the temperature range of 20–40 °C (imposed by the thermal stability of the METASORB solution), the sorption capacity of Cu(II) ions varies quite a bit (from 0.18 mmol/g at 20 °C to 0.22 mmol/g at 40 °C). Under these conditions, the temperature of 30 °C was chosen as optimal for functionalization, because it keeps all the advantages offered by the temperature increase without bringing unjustified additional costs (Figure 2b).

Based on the experimental results presented in Figure 2, the functionalization of MWB with METASORB is achieved with maximum efficiency at a mixture ratio of 1 g MWB: 1.0 mL of METASORB solution and at a temperature of 30 °C. The sorbent obtained under these experimental conditions was named MET-MWB, and this notation will be used hereafter.

To test the stability of MET-MWB in an aqueous solution, 0.5 g of sorbent was mixed with 25 mL of Cu(II) ion solution (0.8 mmol/L, pH = 5.5) for 24 h. The value of the COD index (mg O_2_/L) determined for the aqueous solution before and after the sorbent separation varies insignificantly (from 38.13 mg O_2_/L to 37.95 mg O_2_/L). This shows that the METALSORB molecules are bonded to the MWB surface by strong interactions that prevent their release when in contact with the aqueous solution. Therefore, the obtained sorbent is stable and can be used in the metal ion retention processes.

### 3.2. Characterization of MET-MWB

In order to highlight the structural features of MET-MWB necessary for its use as a sorbent, FTIR spectra and SEM images were recorded (Figure 3 and Figure 4).

As can be seen from Figure 3, after the functionalization of MWB, some significant changes can be noted. Thus, the broad band at 3300–3435 cm^−1^ in MWB (spectra 1), corresponding to the stretching vibrations of O–H and N–H bonds, becomes much sharper and shows a single absorption maximum (3435 cm^−1^) in MET-MWB (spectra 3), suggesting a rearrangement of the –OH and –NH_2_ groups. Similar changes can be observed in the case of the bands at 1544–1743 cm^−1^ and 997–1105 cm^−1^ (spectra 1), which correspond to the stretching vibrations of C=O bonds from carbonyl and carboxyl compounds, and, respectively, C–O bonds from oxygenated compounds. The increase in the intensity of these bands and the reduction in the number of absorption maxima after functionalization (spectra 3) shows that these groups contribute to the retention of METALSORB molecules on MWB, most likely through hydrogen bonds. On the other hand, some new bands appear in the MET-MWB spectrum (spectra 3), which are characteristic of METALSORB (spectra 2). Thus, the bands at 2061 and 1319 cm^−1^ are characteristic of the stretching vibrations of C–N and C=S bonds in saturated amines and thiols. In addition, the band at 1415 cm^−1^ indicates the presence of N–C=S groups. Based on these observations, it can be said that the functionalization of MWB was successfully achieved, and the obtained sorbent has a series of new functional groups on its surface, which can increase its efficiency in the retention of metal ions from aqueous media.

The differences observed in the FTIR spectra of MWB before and after functionalization (Figure 3) are also supported by the changes in the surface morphology of the two materials, as recorded by the SEM images (Figure 4). Thus, it can be observed that, after functionalization, new non-uniform formations appear on the MET-MWB surface (Figure 4b).

These formations are most likely due to METALSORB macromolecules that form local conglomerates, and their presence leads to an increase in the roughness of the new sorbent, and thus to an increase in its specific surface area. These structural features significantly influence the sorption process [36], and therefore, it is expected that the removal of metal ions will be more efficient when using the functionalized sorbent.

### 3.3. Testing the Sorptive Performances of MET-MWB

To test the performance of the MET-MWB sorbent, three metal ions (Cu(II), Zn(II) and Co(II)) were selected based on their physico-chemical characteristics and their importance in industrial activities. Batch sorption experiments were performed under optimal experimental conditions (initial solution pH = 5.0, sorbent dose = 5.0 g/L; temperature = 21 °C), established in a previous study [29].

#### 3.3.1. Influence of Initial Metal Ion Concentration and Isotherm Modeling

One of the most important parameters influencing the performance of a sorbent is the initial concentration of metal ions. This is because the study of the variation in the sorption efficiency as a function of the initial metal ion concentration allows one to obtain essential information for establishing: (i) the concentration range of metal ions in which the sorbent allows their quantitative removal; and (ii) the maximum concentration of metal ions in which the sorbent saturation occurs [20]. These aspects are important in designing such a system for large-scale applications.

In this study, the initial concentration of metal ions varied in a range between 0.2 and 3.2 mmol/L, and the variation of the sorption capacity (q, mmol/g) for each metal ion is presented in Figure 5a.

The increase in the initial concentration of metal ions (c_0_, mmol/L) causes an increase in the values of the sorption capacity (q, mmol/g) for the entire studied concentration range and for all metal ions (Figure 5a). Such a variation is characteristic of sorption processes [20,21] and is due to the fact that, with the increase in the initial concentration, more and more metal ions reached the surface of the sorbent, where they interact with the superficial functional groups and are retained. Thus, according to the experimental data presented in Figure 5a, increasing the initial metal ion concentration from 0.2 to 3.2 mmol/L leads to an increase in the sorption capacity from 0.01 to 0.42 mmol/g in the case of Cu(II), from 0.02 to 0.29 mmol/g in the case of Zn(II) and from 0.04 to 0.47 mmol/g in the case of Co(II). In this concentration range, the removal percents varied between 29 and 78% for Cu(II), 42 and 65% for Zn(II) and 54 and 91% for Co(II) (data not shown).

Although the increase in sorption capacity is significant in the studied concentration range, from a practical point of view, two aspects merit attention. The first is related to the fact that the sorbent saturation was reached at the high initial concentration of metal ions. As can be seen from Figure 5a, the increase in the sorption capacity is only slower at concentrations higher than 2.4 mmol M(II)/L, although the appearance of a plateau is not obvious. This suggests that MET-MWB allows the treatment of large volumes of effluents in which the metal ion content can vary within fairly wide limits without the need to replace it (due to exhaustion).

The second aspect is related to the efficiency of this sorbent in the treatment of effluents. Although the values of the sorption capacity increase in the order of Co(II) > Cu(II) > Zn(II) (Figure 5a), the graphical representation of the metal ion concentration before (*c*_0_, mmol/L) and after (*c*, mmol/L) the sorption process (Figure 6) indicates a moderate efficiency for MET-MWB.

The regression analysis of the dependencies illustrated in Figure 6 shows that, for the treated effluent to meet the conditions required by NTPA [37], the initial concentration of metal ions must not be higher than 10 mg M(II)/L. This is not viable from a practical point of view. Therefore, it is recommended that MET-MWB is used to reduce the content of metal ions in industrial effluents, and then, the treated effluents are subject to either a new sorption step (with the clean MET-MWB) or another advanced treatment method.

The quantitative evaluation of the sorption of Cu(II), Zn(II) and Co(II) ions on MET-MWB was performed by the mathematical analysis of the equilibrium data using the Langmuir, Freundlich and Temkin isotherm models. The linear representations of these models (according to Equations (4)–(6)) are illustrated in Figure 5b–d, respectively, and the isotherm parameters calculated for each metal ion are summarized in Table 1.

Comparing the regression coefficients (R^2^) obtained for the three isotherm models (Figure 5b–d), it can be seen that the experimental data are best described by the Langmuir model. This means that the retention of Cu(II), Zn(II) and Co(II) ions on MET-MWB is a monolayer sorption process, which takes place on a heterogeneous surface (as demonstrated by the high values of R^2^ in the case of the other two models). The maximum sorption capacities (*q_max_*, mmol/g) (Table 1), calculated from the Langmuir model, have values close to the experimental ones, which explains the flattening of the experimental isotherms at the high initial concentration of metals ions (Figure 5a). In addition, such behavior shows that the retention of metal ions is achieved through the interactions occurring at the surface of the MET-MWB. These interactions are predominantly electrostatic (according to the values of the *B* parameter of the Temkin model) and involve superficial active sites, which makes the sorption process a favorable event at high metal ion concentrations (values of n in the Freundlich model are greater than unity). In addition, the variation in the maximum sorption capacity, which increases in the order of Co(II) > Cu(II) > Zn(II), is mainly determined by the physico-chemical characteristics of the metal ions. Thus, the greater the ratio between the Pauling electronegativity and ionic radius (0.0149 for Co(II), 0.0141 for Cu(II) and 0.0135 for Zn(II)), the more effectively the metal ions interact with the functional groups of the sorbent, and the values of the sorption capacity are greater.

The values of the maximum sorption capacity obtained for the retention of Cu(II), Zn(II) and Co(II) ions on MET-MWB are comparable to the values reported in the literature when using different naturally functionalized materials as sorbents under similar experimental conditions [38,39,40]. However, much more important is the fact that, after functionalization with METALSORB, the maximum sorption capacity of the obtained sorbent (MET-MWB) significantly increased compared with the non-functionalized biomass (MWB). Thus, in the case of Cu(II), this increase is 58.59% (from 0.26 to 0.42 mmol/g), 18.86% in the case of Zn(II) (from 0.24 to 0.29 mmol/g) and 69.35% in the case of Co(II) (from 0.27 to 0.47 mmol/g) (data not shown). All these observations show that MET-MWB has the potential and can be practically used as a sorbent in the decontamination of industrial effluents.

#### 3.3.2. Influence of Contact Time and Kinetic Modeling

In order to determine the minimum contact time required to reach equilibrium, the variation in the sorption capacity at different contact time values (between 5 and 180 min) was examined. Figure 7a illustrates the influence of the contact time on the sorption efficiencies of the Cu(II), Zn(II) and Co(II) ions on MET-MWB.

The variation of the sorption capacity over the entire studied contact time interval (Figure 7a) shows a significant increase in this parameter in the interval of 0–30 min. At higher contact time values, the increase in the sorption capacity of MET-MWB is much slower, indicating that the sorption processes have reached equilibrium. At a contact time of 30 min, the retention of the metal ions is quantitative (68.18% for Cu(II), 46.51% for Zn(II) and 73.84% for Co(II)). The subsequent increase in the contact time (up to 180 min), does not significantly change the values of the retention percentage of metal ions (4.39% for Cu(II), 8.51% for Zn(II) and 2.75% for Co(II)). Therefore, a contact time of at least 30 min ensures that the equilibrium state is reached in the studied sorption processes, and this value once again underlines the practical potential of using MET-MWB in the decontamination processes of industrial effluents.

The modeling the experimental kinetic data was performed using a pseudo-first-order model, pseudo-second-order model and an intra-particle diffusion model (Equations (7)–(9)), and the best-fitting model was selected based on the regression coefficient (R^2^). The linear representations of these three kinetic models are illustrated in Figure 7b–d, while the values of the kinetics parameters are summarized in Table 2.

As can be seen from Figure 7b–d, all kinetic models describe the experimental data very well, as the values of R^2^ are greater than 0.9 in all cases. However, in the case of the pseudo-second-order kinetic model, the calculated values of the sorption capacity (*q_e_*^calc^, mmol/g) are closer to the experimental ones (*q_e_*^exp^, mmol/g) than in the case of the pseudo-first-order model. This allows us to say that the pseudo-second-order model is the most suitable for the analysis of experimental data. Consequently, the retention of Cu(II), Zn(II) and Co(II) ions on MET-MWB is performed by electrostatic interactions that require two binding centers, with a favorable geometric orientation, located on the surface of the sorbent. The rate constants of the pseudo-second-order kinetic model (k_2_, g/mmol min) follow the order of Co(II) > Cu(II) > Zn(II), which is similar to the ratio between electronegativity and ionic radius (0.0149 for Co(II), 0.0141 for Cu(II) and 0.0135 for Zn(II)). This variation in the rate constants indicates that Co(II) ions have a higher affinity for the superficial functional groups of MET-MWB compared with Cu(II) and Zn(II) ions, and this difference can be easily observed in Figure 7a.

However, the two following aspects should also be mentioned. The first is that the pseudo-first-order kinetic model fits the experimental data very well at the low values of the contact time and for all metal ions (Figure 7b). This indicates that, in the initial moments of the sorption processes, when a large number of metal ions reach the surface of the sorbent, their binding occurs through a single-active center. Only after that is the superficial complex stabilized by removing one more water molecule and forming a new bond with another superficial functional group of the sorbent. This behavior shows that the functional groups on the sorbent have high affinity for the metal ions in the aqueous solution, and explains the good correlation between the experimental data and the Freundlich isotherm model (see Section 3.3.1). The second aspect is related to the importance of elementary diffusion processes. As can be seen from Figure 7d, none of the linear representations of the intra-particle diffusion model pass through the origin. This means that elementary diffusion processes are not the rate-controlling step in the retention of Cu(II), Zn(II) and Co(II) ions on MET-MWB. Analyzing the values of the kinetic parameters of the intra-particle diffusion model (Table 2), it can be seen that the concentrations of metal ions in the diffusion layer (*c*_1_, mmol/L) are close to the values of their concentration in the bulk of the solution (*c*_2_, mmol/L), while there are significant differences (*k*_1_ >> *k*_2_) between the rate constants. These observations show that, as soon as the metal ions reach the surface of the sorbent, they find functional groups available for their binding and, consequently, the elementary diffusion steps occur with high rates.

### 3.4. Sorbent Regeneration and Desorption of Metal Ions

To evaluate the regeneration and reuse potential of the sorbent, samples of 0.25 g of MET-MWB loaded with each metal ion were treated for 3 h with 10 mL of HNO_3_ solution (10^−1^ mol/L). After washing (until a neutral pH), the same amount of sorbent was used for another sorption cycle.

The sorption efficiency (R, %) was found to decrease quite a bit in higher cycles compared to the initial value (Figure 8a). Thus, from the first to the third cycle, the R values decrease by 12% in the case of Co(II) ions (from 95% to 83%), by 16% in the case of Cu(II) ions (from 90% to 74%) and by 25% in the case of Zn(II) ions (from 86% to 61%). A similar behavior can be observed in the case of the desorption of metal ions retained on the MET-MWB sorbent. The desorption efficiency (DE, %) decreases by 10% for Co(II) and Zn(II) ions and by 12% for Cu(II) ions with the increasing of number of sorption/desorption cycles (Figure 8b). This decrease in sorption/desorption efficiency in subsequent cycles is most likely determined by the changes occurring at the sorbent surface (ex. inactivation/loss of superficial functional groups) during the regeneration and washing steps. However, the maintenance of good values of metal ion removal percents (R%) up to three cycles, and the high efficiency of metal ion recovery after each cycle show that MET-MWB is a reusable material that can be used in the decontamination processes of industrial effluents.

### 3.5. Sorption Mechanism

The retention of metal ions on the surface of solid materials can be achieved by three distinct mechanisms: ion exchange, superficial complexation and superficial precipitation [16]. The identification of the predominant mechanism is important because it allows one to establish the limits of the use of the sorption process (efficiency in metal ion removal, selectivity, desorption conditions, etc.) in large-scale practical applications.

In the case of using MET-MWB for the retention of Cu(II), Zn(II) and Co(II) ions, the experimental working conditions (pH = 5.0, 5 g sorbent/L) practically exclude the possibility of metal ion precipitation on the sorbent surface. Therefore, ion exchange and superficial complexation remain the only viable options.

The modeling of the kinetic and equilibrium data, presented in the previous sections, revealed that: (i) metal ions are retained on the sorbent surface until a monolayer coverage is formed (according to Langmuir model assumptions); (ii) the maximum sorption capacity depends on the ratio between the electronegativity and the ionic radius of the metal ions; (iii) the sorption energies have relatively low values; (iv) the retention of metal ions requires the existence of two binding centers located in favorable geometric positions (according with pseudo-second-order kinetic model assumptions); and (v) the desorption of metal ions is quantitative in strong acid media.

All these observations indicate that the sorption of Cu(II), Zn(II) and Co(II) ions on MET-MWB is governed by a complex mechanism, involving both electrostatic and donor–acceptor interactions. A schematic representation of the sorption mechanism of metal ions on MET-MWB is illustrated in Figure 9.

The metal ions (positively charged) are attracted to the surface of the sorbent by the negatively charged ionized thiol groups (step 1). These electrostatic interactions most likely represent the first stage of the sorption mechanism, since their realization involves low binding energies, and thus high rates. After the neutralization of the first positive charge of the metal ion, the intermediate complex is stabilized by the formation of a donor–acceptor bond between the metal ion (still positively charged) and the lone pair of electrons of the doubly bonded sulphur atom to carbon (step 2). These donor–acceptor interactions take a little longer to achieve, because the metal ion must find the C=S bond that has a favorable geometric position. It is most likely that both electrostatic and donor–acceptor interactions involve the participation of the same METALSORB molecule; however, the experimental data do not exclude the possibility of the participation of two molecules of the functionalization agent. The involvement of the non-participating electrons of the S atom (C=S bond) in the donor–acceptor interactions with the metal ion, causes a displacement of electrons in the METALSORB molecule due to the electronic effects. Consequently, the electron density on the N atom decreases, and this is essential for the functionalization of MWB, as it allows for the stabilization of METASORB molecules on the biomass surface trough electrostatic interactions and hydrogen bonds (step 3).

All these observations are supported by FTIR spectra and SEM images recorded for MET-MWB before and after metal ion retention. For example, Figure 10 shows the FTIR spectra and SEM images before and after the sorption of Co(II) ions (which are most efficiently retained on MET-MWB); however, similar results were also obtained for Cu(II) and Zn(II) ions.

The changes in the wave numbers corresponding to the absorption bands in the region 970–1155 cm^−1^ and 1317–1732 cm^−1^ (Figure 10a, spectrum 1) show that the functional groups of alcohol and carbonyl type (from MWB) and thiol groups (characteristic of METALSORB) are involved in the retention of metal ions. In addition, the appearance of absorption bands in the 2104–2356 cm^−1^ region (Figure 10a, spectrum 2), which are determined by the stretching vibrations of the C-N bond in tertiary amine salts, supports the hypothesis of METASORB stabilization on the biomass surface through direct interactions between the N atom and biomass functional groups. This hypothesis is also supported by the SEM images (Figure 10b), in which the ordering of the METASORB molecules can be observed after the retention of metal ions.

These observations once again demonstrate that the use of METASORB for MWB functionalization allows one to obtain a sorbent (MET-MWB) with superior performances in the removal of metal ions (as demonstrated in the previous sections), which has potential to be used in large-scale decontamination processes.

### 3.6. Removal of Metal Ions in Wastewater

To test the potential of MET-MWB in the treatment of industrial effluents, the sorption of Cu(II), Zn(II) and Co(II) ions was performed in a single-component system, using wastewater samples (purchased from local metal coating company). In each wastewater sample, the initial concentration of metal ions was adjusted at 0.8 mmol/L (~50 mg/L), and the sorption experiments were performed under the following conditions: initial pH of 5.0, 0.5 g of MET-MWB, contact time of 3 h and temperature of 21 °C. The concentration of each metal ion, as well as the values of some quality parameters used for the evaluation of wastewater [37], before and after sorption on MET-MWB, are presented in Table 3.

More than 95% of the metal ion content (97.54% for Cu(II), 96.80% for Zn(II) and 95.68% for Co(II)) is removed from wastewater samples using MET-MWB. Although the sorption capacities obtained in wastewater samples are lower than those obtained in laboratory solutions (±10–15%), and the concentrations of metal ions remaining after sorption are higher than the maximum permissible values (see Table 3), MET-MWB could effectively remove metal ions from industrial effluents. In addition, the values of other quality parameters (COD, water hardness, turbidity, electrical conductibility) do not change significantly (Table 3), indicating that MET-MWB is a chemically stable material that behaves similarly to an ion exchanger.

## 4. Conclusions

In this study, mustard waste biomass (obtained after oil extraction) was functionalized with METALSORB (an industrial polymeric thiocarbamate) and used as a sorbent for the removal of Cu(II), Zn(II) and Co(II) ions, in batch systems. The functionalization conditions were: mixing ratio biomass: METASORB = 1 g: 1.0 mL and a temperature of 30 °C, and the obtained sorbent (MET-MWB) has the potential to be used as a sorbent for the removal of metal ions. The equilibrium data best fit the Langmuir model, and the maximum sorption capacity increase in the order of Zn(II) (0.24 mmol/g) < Cu(II) (0.41 mmol/g) < Co(II) (0.43 mmol/g). The pseudo-second-order model best explains the sorption kinetics, although elementary diffusion steps significantly contribute to the realization of sorption processes. The predominant interactions between MET-MWB and metal ions are of the electrostatic and superficial complexation type, and this is supported by the FTIR spectra. The sorbent can be reused in up to three sorption/desorption cycles without significant changes in its efficiency. In addition, MET-MWB is an good and cost-effective sorbent for the removal of metal ions (Cu(II), Zn(II) and Co(II)) from real wastewater samples.

## Figures and Tables

**Figure 1 polymers-15-02301-f001:**
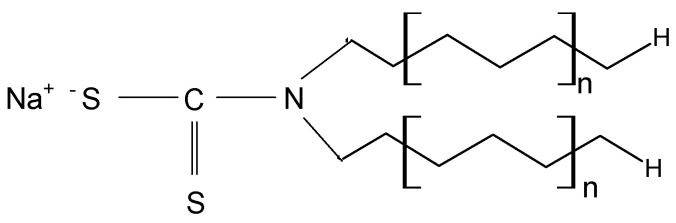
Chemical structure of METALSORB (n is the number of ethylene groups from the aliphatic chain).

**Figure 2 polymers-15-02301-f002:**
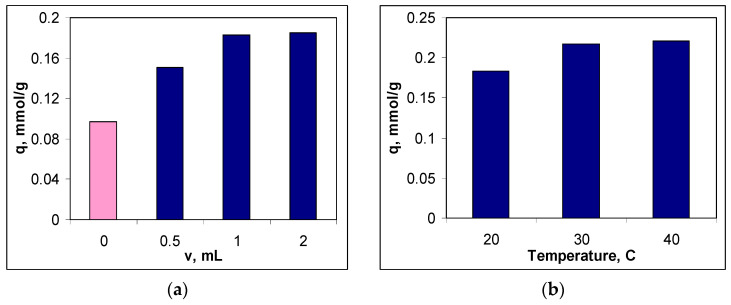
Influence of the volume of METALSORB solution (20 °C) (**a**) and the temperature (1.0 mL of METASORB) (**b**) on the sorption capacity of Cu(II) ions and on the functionalized sorbent (pH = 5.5; 5.0 g sorbent/L; *c*_0_ = 0.8 mmol/L, 3 h, 20 °C).

**Figure 3 polymers-15-02301-f003:**
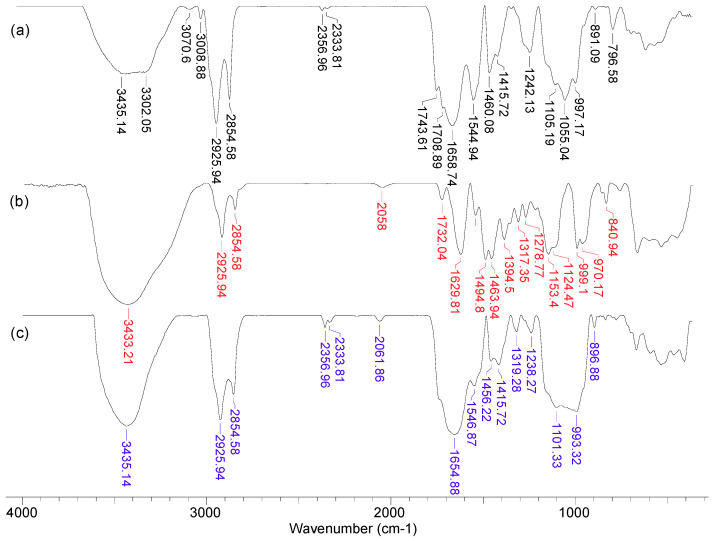
FTIR spectra of MWB (**a**), METALSORB (**b**) and MET-MWB (**c**).

**Figure 4 polymers-15-02301-f004:**
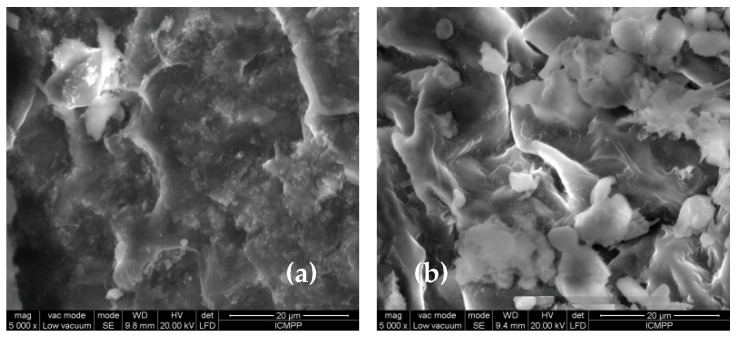
SEM images of MWB (**a**) and MET-MWB (**b**) at an order of magnitude of 5000×.

**Figure 5 polymers-15-02301-f005:**
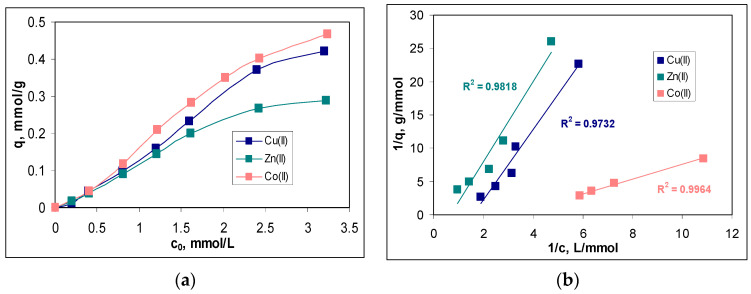
Variation of the sorption capacity as a function of the initial concentration of metal ions (**a**) and the linear representations of the Langmuir (**b**), Freundlich (**c**) and Temkin (**d**) models for the sorption of Cu(II), Zn(II) and Co(II) on MET-MWB.

**Figure 6 polymers-15-02301-f006:**
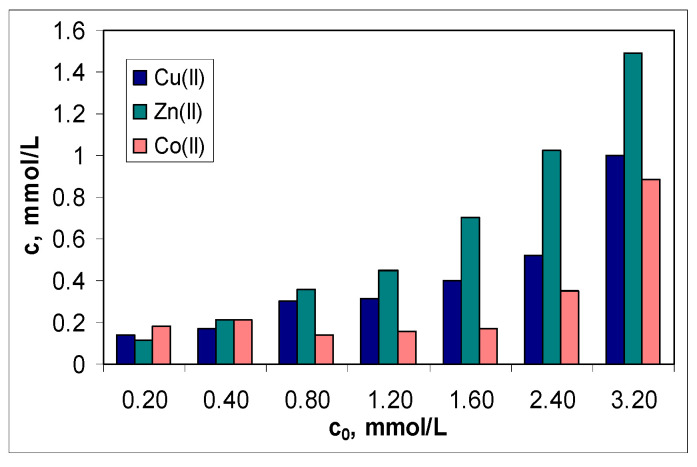
*c* vs. *c*_0_ for the sorption of Cu(II), Zn(II) and Co(II) ions on MET-MWB.

**Figure 7 polymers-15-02301-f007:**
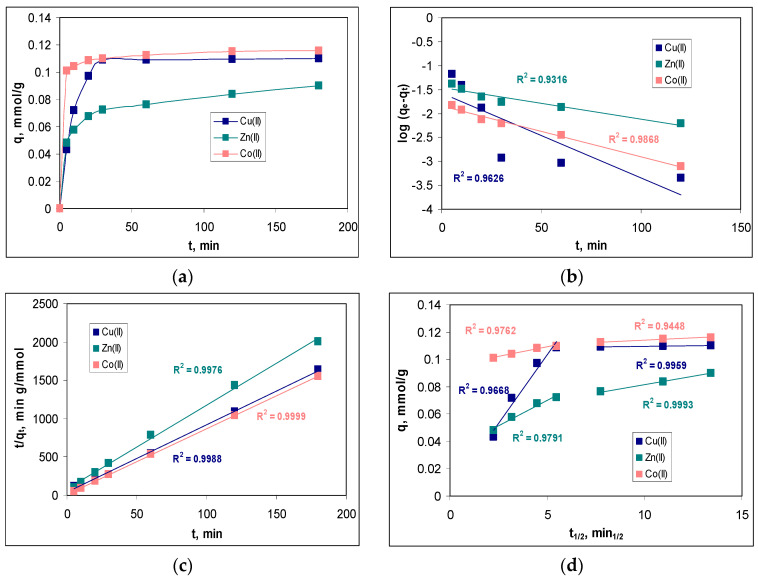
Effect of the contact time (**a**) and linear representations of the pseudo-first-order model (**b**); pseudo-second-order model (**c**) and intra-particle diffusion model (**d**) for the sorption of Cu(II), Zn(II) and Co(II) on MET-MWB.

**Figure 8 polymers-15-02301-f008:**
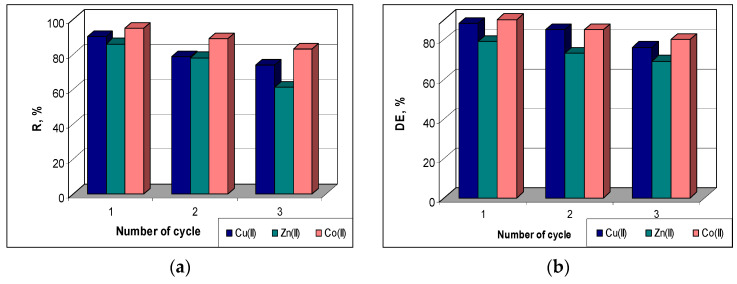
Sorption efficiency (R, %) (**a**) and desorption efficiency (DE, %) (**b**) of Cu(II), Zn(II) and Co(II) ions on MET-MWB.

**Figure 9 polymers-15-02301-f009:**
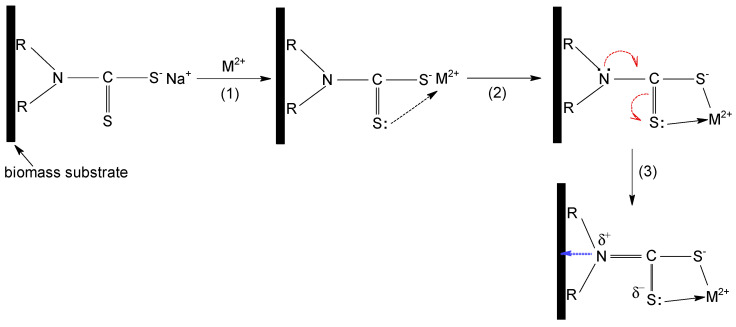
Schematic representation of the sorption mechanism of Cu(II), Zn(II) and Co(II) ions on MET-MWB.

**Figure 10 polymers-15-02301-f010:**
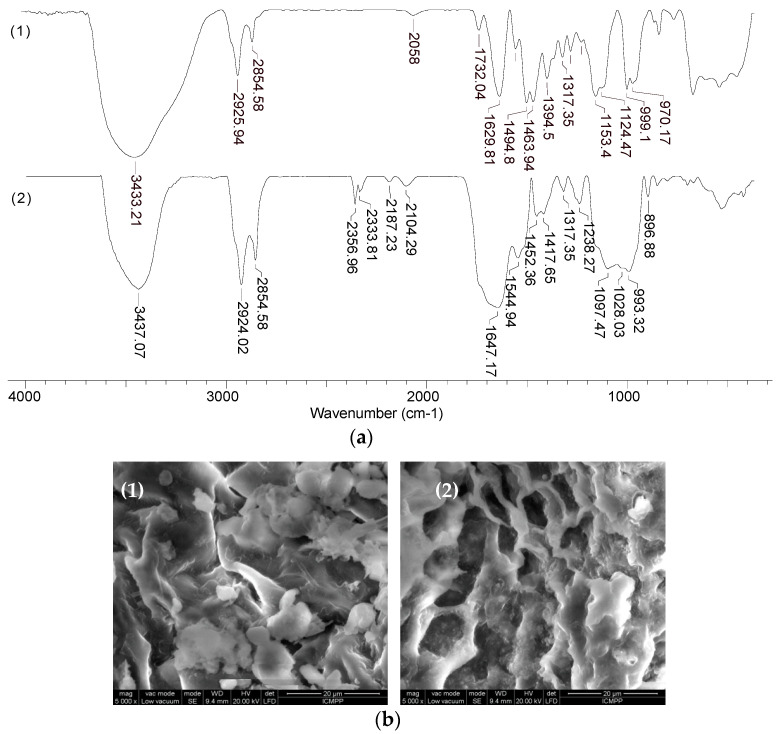
FTIR spectra (**a**) and SEM images (**b**) of MET-MWB before (**1**) and after (**2**) Co(II) ion sorption.

**Table 1 polymers-15-02301-t001:** Isotherm parameters for Cu(II), Zn(II) and Co(II) ion sorption on MET-MWB.

Isotherm Parameter		Cu(II)	Zn(II)	Co(II)
Langmuir model	*q_max_*, mmol/g *K_L_*, L/mmol	0.4106 1.5867	0.2399 0.6904	0.4282 3.1635
Freundlich model	n *K_F_*, L^1/n^/g·mmol^1/(n−1)^	2.62 0.6076	1.98 0.2477	3.58 2.2979
Temkin model	*A_T_*, L/g *B*, kJ/mol	6.7715 22.96	5.7959 11.80	8.2599 24.83

**Table 2 polymers-15-02301-t002:** Kinetic parameters for the Cu(II), Zn(II) and Co(II) ion sorption on MET-MWB.

Kinetic Parameter		Cu(II)	Zn(II)	Co(II)
*q_e_*^exp^, mmol/g		0.1102	0.0898	0.1159
Pseudo-first-order model	*q_e_*^calc^, mmol/g k_1_, 1/min	0.0265 0.0177	0.0347 0.0066	0.0144 0.0106
Pseudo-second-order model	*q_e_*^calc^, mmol/g k_2_, g/mmol min	0.1136 1.9952	0.0916 1.4588	0.1166 5.8479
Intra-particle diffusion model	*c*_1_, mmol/L k_diff_^1^, mmol/g min^1/2^	0.0933 0.0201	0.0327 0.0057	0.0951 0.0028
	*c*_2_, mmol/L	0.1081	0.0578	0.1077
	k_diff_^2^, mmol/g min^1/2^	0.0002	0.0024	0.0006

**Table 3 polymers-15-02301-t003:** Quality parameters of wastewater before and after treatment with MET-MWB.

Parameter	Initial Wastewater	Treated Wastewater	Maximum Permissible Value [37]
Cu(II), mg/L	50.83	1.25	0.2
Zn(II), mg/L	52.31	1.67	1.0
Co(II), mg/L	47.14	0.61	1.0
pH *	5.0	6.12	6.5–8.5
CCO, mg O_2_/L *	318.43	275.14	500
Water hardness, ° Ge *	13.45	13.79	-
Turbidity, NTU *	11.02	7.13	-
Electrical conductibility, μS/cm *	1327	1219	-

* Average values for three samples.

## Data Availability

Not applicable.

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
