# Peer review of "Isotherm and Kinetic Study of Metal Ions Sorption on Mustard Waste Biomass Functionalized with Polymeric Thiocarbamate"

_polymers, 2023, doi:10.3390/polym15102301_

Round 1
Reviewer 1 Report
1. The quality of the language needs to be upgraded. Some spelling mistakes are noticed in the manuscript.
2. Lines 119 & 305. The abbreviation for chemical oxygen demand should be 'COD', not CCO.
3. Methodology section - Why did the authors not carry out the BET surface area and pore volume analysis on the synthesized adsorbent?
4. Section 2.3. It is not clear whether the metal adsorption experiment was carried out in a single metal system or a ternary system. Please state in the methodology section.
5. Section 2.5. As real wastewater was used, does this mean the authors performed the metal ion analysis using AAS by standard addition method? Please explain.
6. Lines 247-261. These two paragraphs are more suitable to be placed in the Introduction section.
7. Figure 2 (a & b). Please use proper symbols that represent volume and temperature (oC)
8. Line 363. Figure 5a did not represent the % Removal of metal ions.
9. Lines 429-431. This statement requires a reference, especially the ionic radius and electronegativity of each metal ion. Did the author mean Pauling electronegativity?
10. Figure 10. Where are the EDX images of the MET-WEB (before and after metal adsorption)? Also, where are the FTIR spectra of MET-WEB (after Zn and Cu adsorption)? Only MET-WEB + Co(II) spectrum was presented.
11. Table 3. Why is there an increase in the water hardness value in the treated wastewater sample?
12. Authors stated that chemical sorption (complexation) and electrostatic attraction were responsible for the metal ions removal. Which one is dominant?
Please send this manuscript for proofreading.
Author Response
Dear Reviewer,
Thank you very much for your observations and recommendations. We read them very carefully and all have been considered in this variant of manuscript. All corrections were marked with yellow to make it easier to observe.
- The quality of the language has been improved. I hope that all mistakes have been corrected.
- Lines 119 and 305: The abbreviation has been corrected.
- Methodology section: No, because in this case the determination of pore volume and BET surface area are less important. Our experimental studies were aimed at finding conditions for the synthesis of this adsorbent that would ensure a high adsorption capacity and reasonable stability.
- Section 2.3: The metal adsorption experiments were performed in mono-component systems. This aspect is now clearly mentioned in this section.
- Section 2.5: For the analysis of real wastewater, the metal ion analysis was performed by AAS using a prepared calibration curve. This is now mentioned in the manuscript.
- Lines 247-261: These paragraphs are just a brief presentation of the most important key factors of this study. In our opinion, it should stay here.
- Figure 2 (a and b): This correction has been made.
- Line 363: I agree. This correction has been made.
- Lines 429-431: Yes, it is Pauling electronegativity. This is now clear mentioned in the manuscript.
- Figure 10: In this figure we included, for exemplification, only the FTIR spectra and SEM images for MET-MWB, before and after Co(II) ions sorption, because in this case the higher sorption capacity was obtained. For the other two ions (Cu(II) and Zn(II)), the shape of FTIR spectra and SEM images were similar, and we considered their addition to bring nothing new to the interpretation of the mechanism.
- Table 3: There is no increase in water hardness after the treatment. The difference (0.34 °Ge) is irrelevant.
- It’s hard to say. But, the experimental data show that the binding of metal ions involves electrostatic interactions that lead to surface complexation.
I hope that all these details are sufficiently to clarify the aspects underlined in the reviewer comments.
Best regards,
Prof.dr.habil.chem. Laura Bulgariu
Department of Environmental Engineering and Management
“Cristofor Simionescu” Faculty of Chemical Engineering and Environmental Protection
Technical University “Gheorghe Asachi” of IaÅŸi
D. Mangeron 71A, 700050-IaÅŸi, Romania
E-mail: lbulg@ch.tuiasi.ro / laurabul73@yaho.com
Reviewer 2 Report
1. The outer border of the Figure is redundant and it is recommended to delete it.
2. Why did select (II), Zn(II) and Co(II) as the target adsortion ions?
3. How is the repeatability of the results? Perhaps error bars need to be added.
English is good.
Author Response
Dear Reviewer,
Thank you very much for your observations and recommendations. We read them very carefully and all have been considered in this variant of manuscript. All corrections were marked with yellow to make it easier to observe.
- This representation of the figures is preferred by us. Therefore, since the Instructions for authors allow us to do so, we would like the figures to remain in this form.
- We select Cu(II), Zn(II) and Co(II) ions for the experimental studies, because they are the most important contaminants (from the category of heavy metals) in the wastewater of the company where we took the sample.
- All the experimental results included in this study have good repeatability. We have avoided adding errors bars in order not to unnecessarily complicate the figures (most figures include representations of the three metal ions studied).
I hope that all these details are sufficiently to clarify the aspects underlined in the reviewer comments.
Best regards,
Prof.dr.habil.chem. Laura Bulgariu
Department of Environmental Engineering and Management
“Cristofor Simionescu” Faculty of Chemical Engineering and Environmental Protection
Technical University “Gheorghe Asachi” of IaÅŸi
D. Mangeron 71A, 700050-IaÅŸi, Romania
E-mail: lbulg@ch.tuiasi.ro / laurabul73@yaho.com
Reviewer 3 Report
In this article, mustard waste biomass, functionalized with polymeric thiocarbamate, was developed for the sorption of Cu(II), Zn(II) and Co(II) from aqueous solutions. The research topic is really interesting and the manuscript is well-organized. However, at some points, the submitted manuscript leaves something to be desired. Hence, it can be recommended for publication after a major revision considering the following comments:
Comment #1: Abstract should be supported with more qualitative findings (too much general introduction sentences).
Comment #2: The whole manuscript should be revised for words spelling (e.g., Line 108 (fulfill)).
Comment #3: Introduction is well-written. However, the novelty aspects and detailed objectives of this work should be clearly highlighted. Furthermore, the scientific soundness of the context should be enhanced.
Comment #4: Charts should be supported with error bars, whenever possible.
Comment #5: Super/subscript formatting should be rechecked in the whole text.
Comment #6: Sips isotherm model (https://doi.org/10.3389/fnuen.2023.1142823) could exhibit better fitting with the experimental sorption data, as it combines the advantages of both Langmuir and Freundlich models.
Comment #7: Akaike information criterion (AIC) can be more accurate than coefficient of determination (R2) in models fitting (knowing that it considers the sum of squared errors and the model parameters). In this regard, the following reference may help with either citing/reading: https://doi.org/10.3389/fnuen.2023.1142823
Comment #8: It is strongly recommended to include the sorption-time profiles of Cu(II), Zn(II), and Co(II) in real wastewater.
Comment #9: How about the pH effect on the sorption of the metal ions by the proposed bio-polymeric composite? Will it affect the removal pathways?
Word spelling and grammatical mistakes should be checked within the whole manuscript.
Author Response
Dear Reviewer,
Thank you very much for your observations and recommendations. We read them very carefully and all have been considered in this variant of manuscript. All corrections were marked with yellow to make it easier to observe.
- Abstract: More quantitative results have been added in this section.
- The quality of the language has been improved. I hope that all mistakes have been corrected.
- Introduction: This section has been rechecked. I hope that now all the aspects mentioned by you are much better highlighted.
- Charts: We have avoided adding errors bars in order not to unnecessarily complicate the figures (most figures include representations of the three metal ions studied).
- Super-sub-script formatting has been rechecked, and in some cases corrected.
- Thank you for the suggestion. This paper is very interesting. However, we prefer modelling using the Langmuir, Freundlich and Temkin models due to their broad applicability in such cases.
- Thank you again for the suggestion, the paper suggested by you is very useful, but our arguments remain the same as above.
- The addition of a sorption-time profile in the case of the wastewater sample is irrelevant. As mentioned in the manuscript, the contact time between the two phases is 3 h (quite small compared to other examples presented in the literature). But we will consider this suggestion for further studies, where the kinetic analysis will be discussed in much more detail.
- Surely. This is precisely why the manuscript stated that the optimum pH is 5.0. This value was established experimentally (see reference 29).
I hope that all these details are sufficiently to clarify the aspects underlined in the reviewer comments.
Best regards,
Prof.dr.habil.chem. Laura Bulgariu
Department of Environmental Engineering and Management
“Cristofor Simionescu” Faculty of Chemical Engineering and Environmental Protection
Technical University “Gheorghe Asachi” of IaÅŸi
D. Mangeron 71A, 700050-IaÅŸi, Romania
E-mail: lbulg@ch.tuiasi.ro / laurabul73@yaho.com
Round 2
Reviewer 1 Report
Line 3. The word thyocarbamate should be changed to thiocarbamate.
Line 435. The word si should be changed to 'and'.
Author Response
Dear Reviewer,
Both corrections mentioned in your comments have been made.
Best regards,
Prof.dr.habil.chem. Laura Bulgariu
Reviewer 3 Report
The authors have adequately addressed all the comments.
Author Response
Dear Reviewer,
Thank you for your kindness and help in completing this manuscript.
Best regards,
Prof.dr.habil.chem. Laura Bulgariu